# Neomorphic *PDGFRA* extracellular domain driver mutations are resistant to PDGFRA targeted therapies

Carman K.M. Ip[1], Patrick K.S. Ng[2], Kang Jin Jeong[1], S.H. Shao[2], Zhenlin Ju[3], P.G. Leonard[4,5], Xu Hua[1], Christopher P. Vellano[6], Richard Woessner[7], Nidhi Sahni [1,8], Kenneth L. Scott[9] & Gordon B. Mills[1,2]

Activation of platelet-derived growth factor receptor alpha (PDGFRA) by genomic aberrations contributes to tumor progression in several tumor types. In this study, we characterize 16 novel *PDGFRA* mutations identified from different tumor types and identify three previously uncharacterized activating mutations that promote cell survival and proliferation. PDGFRA Y288C, an extracellular domain mutation, is primarily high mannose glycosylated consistent with trapping in the endoplasmic reticulum (ER). Strikingly, PDGFRA Y288C is constitutively dimerized and phosphorylated in the absence of ligand suggesting that trapping in the ER or aberrant glycosylation is sufficient for receptor activation. Importantly, PDGFRA Y288C induces constitutive phosphorylation of Akt, ERK1/2, and STAT3. PDGFRA Y288C is resistant to PDGFR inhibitors but sensitive to PI3K/mTOR and MEK inhibitors consistent with pathway activation results. Our findings further highlight the importance of characterizing functional consequences of individual mutations for precision medicine.

[1] Department of Systems Biology, The University of Texas MD Anderson Cancer Center, 1515 Holcombe Boulevard, Houston, TX 77030, USA. [2] Sheikh Khalifa Bin Zayed Al Nahyan Institute for Personalized Cancer Therapy, The University of Texas MD Anderson Cancer Center, 1515 Holcombe Boulevard, Houston, TX 77030, USA. [3] Department of Bioinformatics and Computational Biology, The University of Texas MD Anderson Cancer Center, 1515 Holcombe Boulevard, Houston, TX 77030, USA. [4] Institute for Applied Cancer Science, The University of Texas MD Anderson Cancer Center, 1881 East Road, Houston, TX 77054, USA. [5] Core for Biomolecular Structure and Function, Department of Genomic Medicine, The University of Texas MD Anderson Cancer Center, 1881 East Road, Houston, TX 77054, USA. [6] Center for Co-Clinical Trials, The University of Texas MD Anderson Cancer Center, Houston, TX 77030, USA. [7] Cancer Bioscience, in vivo Cancer Pharmacology, AstraZeneca Phamaceuticals, Boston, MA 02451, USA. [8] Department of Epigenetics and Molecular Carcinogenesis, The University of Texas MD Anderson Cancer Center, 1808 Park Rd 1C, Smithville, TX 78957, USA. [9] Dan L. Duncan Cancer Center, Baylor College of Medicine, One Baylor Plaza, Suite 450A, Houston, TX 77030, USA. Correspondence and requests for materials should be addressed to C.K.M.I. (email: ckip@bsd.uchicago.edu)

Cancers are driven by genomic aberrations that activate one or more signaling pathways. The implementation of next-generation sequencing has facilitated the identification of the spectrum of genomic aberrations in tumors from specific patients. One of the key challenges for precision oncology is determining the functional consequences of each mutation and whether they are drivers or passengers arising from underlying genomic instability. Furthermore, neomorphic mutations drive tumor progression through pathways distinct from that parent molecule resulting in altered sensitivity to targeted therapeutics[1]. In-depth functional characterization of the broad spectrum of mutations present in cancer patients, therefore, is critical for precision oncology.

The platelet-derived growth factor (PDGF) receptor is a class III receptor tyrosine kinase, with five extracellular N-glycosylated immunoglobulin (Ig)-like domains, a single transmembrane domain, and a split intracellular protein tyrosine kinase domain. Following binding of PDGF to the extracellular Ig-like domains 2 and 3, the receptors undergo homo- or hetero-dimerization mediated by Ig-like domain 4 inducing a conformation change that leads to autophosphorylation of specific intracellular tyrosine residues. The phosphorylated tyrosine residues create docking sites recruiting a suite of linker molecules that activate the downstream signaling pathways promoting cellular events including proliferation, survival, migration, and differentiation[2]. The spectrum of phosphotyrosines on the activated PDGFR determines the patterns of downstream signals activated and thus functional outcomes.

Elevated PDGFR signaling through either amplification of PDGFs or PDGFR aberrations has been associated with the development and progression of tumors. Activating *PDGFRA* aberrations, including mutation, amplification, and gene fusions including *FLP1L1-PDGFRA*[3], have been described in a number of malignancies. *PDGFRA* mutations are frequently observed in patients with gastrointestinal stromal tumors (GIST; 5–10%)[4], non-small cell lung cancer (6%, TCGA), and colorectal cancer (5%, TCGA). Amplification of *PDGFRA* is a common event in glioblastoma (12%, TCGA), particularly of the proneural subtype[5,6]. Although the overall *PDGFRA* mutation frequency in gliomas is relatively low (1%, TCGA), *PDGFRA* mutations are exclusively observed in rapidly growing high-grade gliomas[7] and tumors with *PDGFRA* amplification (12–40%)[7,8]. Functional characterization of *PDGFRA* mutations have focused on activating mutations in exon 12, 14, and 18 that encode the juxtamembrane and intracellular kinase domains[9–12]. Multiple studies have demonstrated that mutations in extracellular domains of receptor tyrosine kinases, such as fibroblast growth factor receptor 2, colony stimulating factor 1 receptor (*CSF1R*), colony stimulating factor 3 receptor, and epithelial growth factor receptor, confer oncogenic activities[13–16] as well as alter therapeutic liabilities[17]. Given that over half of the point mutations in *PDGFRA* in a number of tumor lineages are located in the extracellular domain (GENIE) and the lack of systematic functional study on missense mutations located in the extracellular domain of PDGFRA, functional characterization of extracellular domain mutations represents a critical need. In this study, we investigate the functional consequences of a series of *PDGFRA* missense mutations and functionally characterize a driver mutation in the extracellular domain of PDGFRA demonstrating that it acts as a neomorphic mutation with important therapeutic implications.

## Results

**PDGFRA Y288C expression enhances malignant phenotypes.** We selected 16 PDGFRA mutations which have not been studied

or characterized from the TCGA, GENIE, or identified in MD Anderson Cancer Center (MDACC) patients across 15 tumor lineages. The selected PDGFRA mutations were mainly from cutaneous melanoma (7 mutations), glioblastoma (6 mutations), and colon/ colorectal adenocarcinoma (5 mutations) (Supplementary Table 1). Eleven out of 16 mutations are recurrent mutations (Supplementary Table 1). Among the 16 selected mutations, 9 were located in the extracellular domain, 2 in the juxtamembrane domain, 3 in the kinase domain, and 2 in the C-terminal domain (Supplementary Table 1). Extracellular domain mutations, T153A, G185W, V193I, and S308F were predicted to be the surface of the 3D structure of the receptor in the dSysMap database[18] (http://dsysmap.irbbarcelona.org/) based on the crystal structure of the extracellular of PDGFRB (PDB code: 3MJG) (Supplementary Table 1). Y288C mutation was predicted to be a buried mutation. The crystal structure of the extracellular domain of PDGFRB was used because of the lack of available crystal structure of the extracellular domain of PDGFRA. The 3D structural location of R340Q, P345S, P414L, S478L, and V484M, were not able to be classified using the dSysMap database. All mutations in the juxtamembrane and kinase domain were classified as surface mutations in the dSysMap database based on the crystal structure of the kinase domain of PDGFRA (PDB code: 5K5X) (Supplementary Table 1). Four of the surface mutations, P577T, E699D, R764C, and G829E, were located at the interface for PDGFRA interactions with EGFR (P577T), PDGFRB (E699D, R746C, and G829E), and PLCG1 (G829E). To study the transforming activity of the *PDGFRA* mutations, we used interleukin-3 (IL-3) dependent Ba/F3, a bone marrow-derived pro-B cell, and EGF/insulin-dependent MCF10A, a non-tumorigenic breast epithelial cell line, as previously described[19]. In our transient transduction assay, two recurrent activating *PDGFRA* mutations, V561D and D842V, in juxtamembrane and kinase domains, respectively, were included as positive controls[9,20]. Extracellular domain mutation S478P, which had previously been shown not to alter functional activity, was included as a negative control[9,21,22]. Among the 16 uncharacterized *PDGFRA* mutations, we identified three novel activating mutations, Y288C, P345S, and P577T (Dunnett's multiple comparisons test; adjusted $p = 0.001$), based on ability to induce cell survival and proliferation of Ba/F3 in the absence of IL-3 to a similar degree to the recurrent constitutively activating PDGFRA mutations, V561D and D842V (Dunnett's multiple comparisons test; adjusted $p = 0.0001$; Fig. 1a). In transient assays in MCF10A in the absence of EGF/insulin, Y288C (Dunnett's multiple comparisons test; adjusted $p = 0.0021$) was activating; however, P345S (Dunnett's multiple comparisons test; adjusted $p = 0.18$) and P577T (Dunnett's multiple comparisons test; adjusted $p = 0.99$) were not in this context (Supplementary Fig 1a). The transformation activity of a mutation was independent of the virus infection rate and the expression level[19]. The activating mutations were exclusively identified from breast carcinoma, GIST, glioma, and melanoma; with glioma having the highest number of activating mutations (Supplementary Fig 1b).

In order to have a better understanding on what features contributed to the functional activities of the mutations, we analyzed and compared the rank score of pathogenicity and evolutionary conservation between activating and non-activating mutations (Supplementary Data 1). The rank score of PRO-VEAN[23], VEST3[24], PolyPhen-2[25], SIFT[26], and CADD[27] were significantly different when compared between activating and non-activating mutations (Supplementary Fig 1c). In addition, all activating mutations were classified as "deleterious" in PRO-VEAN, SIFT and "probably damaging" in PolyPhen-2 (Supplementary Data 1), suggesting that activating mutations would probably lead to a more significant impact on the biological

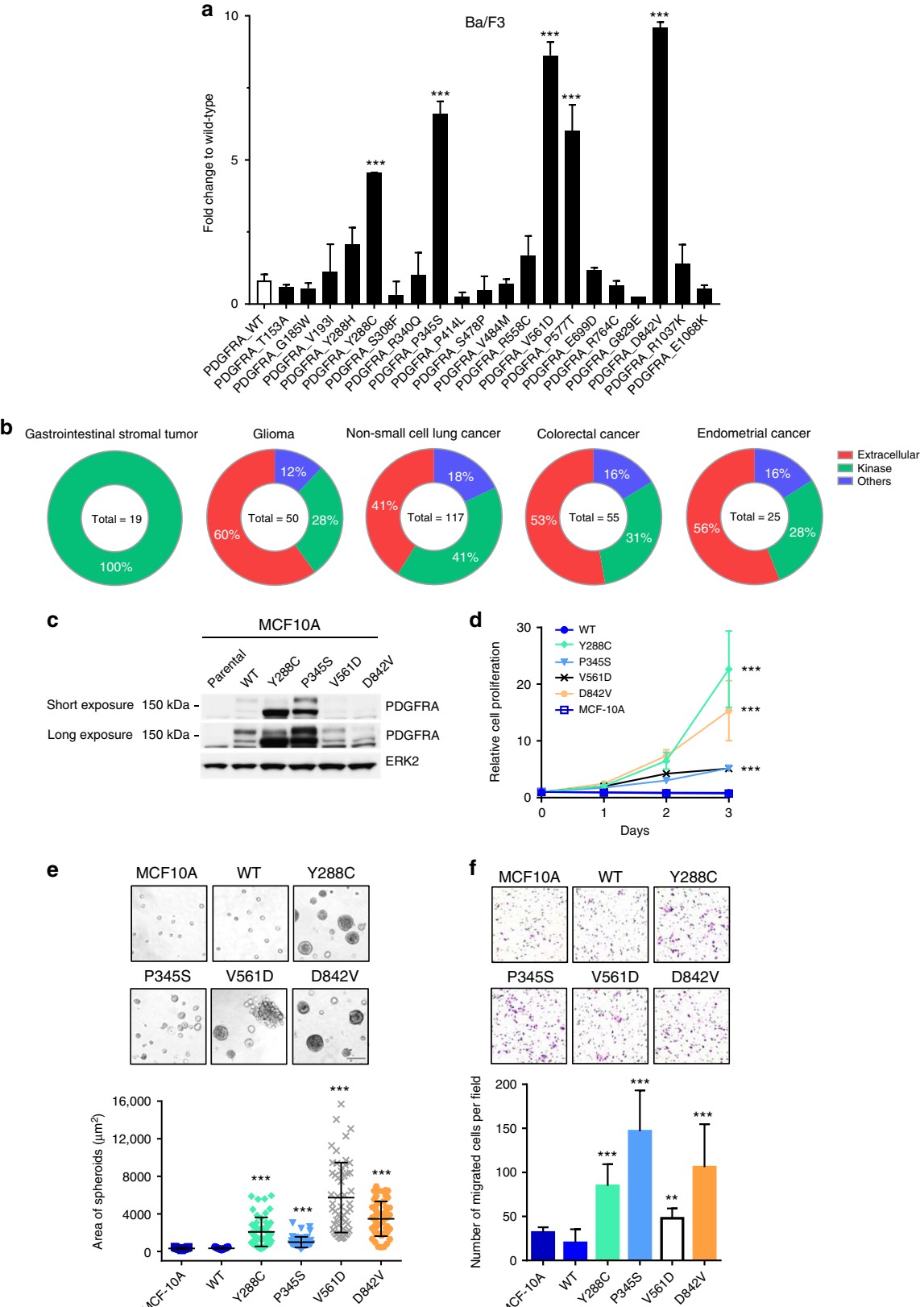

function of the receptor than non-activating mutations. There was no significant difference in the phastCon[28] rank score between the activating and non-activating mutations (Supplementary Fig 1c, bottom right penal), showing that evolutionary conservation was not correlated with the functional activities of

the mutations. Further, based on the 3D structural location, none of the mutations on the surface of the extracellular domain was activating, while 3 out of 7 of surface mutations on the kinase domain were activating (Supplementary Table 1), suggesting that surface mutations on the kinase domain might have a higher

**Fig. 1** Activating PDGFRA mutations differentially promote malignant phenotypes. **a** Functional activity of a series of *PDGFRA* mutations in the transient transfection assay using Ba/F3. The transformation activity of each mutation was tested at least three times. The cell viability in the absence of IL-3 of duplicates was measured 7 days after infection with lentiviral virus carrying PDGFRA WT or mutants and presented relative to that of cells expressing PDGFRA WT (Dunnett's multiple comparisons test). **b** The distribution of *PDGFRA* missense mutations in the GENIE dataset across different domains. Total number of *PDGFRA* missense mutations per tumor type is shown in the middle of each chart. **c** Expression of PDGFRA WT and mutants in MCF10A stable cell lines was examined by western blot. **d** Relative cell proliferation of MCF10A parental and cells stably expressing PDGFRA WT or mutants in the absence of EGF/insulin was measured every 24 h. The cell proliferation of three independent replicates relative to day 0 are presented as mean ± SD (exponential growth equation). **e** MCF10A parental, PDGFRA WT, and mutant expressing cells were grown 3D in growth factor-reduced Matrigel for 10 days. Bar, 100 µm. The size of 50 spheroids are presented as mean ± SD (Dunnett's multiple comparisons test). **f** The cell motility of MCF10A parental, PDGFRA WT, and mutant expressing cells was assessed by Boyden chamber migration assay using 5% serum as chemoattractant for 20 h. The number of migrated cells in five fields of each replicate were quantified and presented as mean ± SD (Dunnett's multiple comparisons test). **p > 0.01; ***p > 0.001

functional impact on receptor function than those on the extracellular domain. Only one of the interface mutation, P577T, was activating (Supplementary Table 1).

P577T on the surface of the juxtamembrane domain was found in only one patient and has not previously been functionally characterized; however, P577S has been shown as a gain-of-function mutation leading to autophosphorylation[29]. Somewhat unexpectedly Y288C and P345S were located in the extracellular immunoglobulin-like domain III and IV of PDGFRA, respectively. In addition to the two glioblastoma samples where Y288C was identified in this study, Y288C has been previously reported in anaplastic oligodendroglioma[30] and recurrently mutated and overexpressed in pediatric high-grade glioma[11,31] while P345S has been identified in breast and melanoma patients in The Cancer Genome Atlas (TCGA) dataset in addition to one glioblastoma patient identified in the MDACC dataset. Using sequencing data from Project GENIE, which contains 453 PDGFRA missense mutations from over 18,000 samples across 37 tumor types, we discovered that over 50% of PDGFRA missense mutations were located in the extracellular domain in glioma and colorectal cancer (Fig. 1b and Supplementary Data 2). Similarly, 50% of the activating mutations we identified were located in the extracellular domain in glioma and none of the activating mutations were located in the extracellular domain in GIST (Supplementary Fig 1b). These data, together with previously identified oncogenic mutations in the extracellular domain of other receptor tyrosine kinases[13–16], suggested the potential contribution of extracellular domain mutations in PDGFRA in cancer pathophysiology. Therefore, we focused our detailed characterization on the Y288C and P345S activating mutations located in the PDGFRA extracellular domain. We decided to use MCF10A and Ba/F3 cells as models based on their responsiveness as well as on their essential lack of endogenous PDGFRA that would complicate interpretation of results in transfected cells (Fig. 1c and Supplementary Fig 2a).

In MCF10A cells stably expressing PDGFRA wild-type (WT) and mutants (Fig. 1c), we found that Y288C strongly promoted cell proliferation, whereas similar to the transient assay, P345S modestly increased proliferation of MCF10A cells in the absence of EGF/insulin (Fig. 1d). Similar effects on viability were observed in Ba/F3 cells stably expressing PDGFRA mutants in the absence of IL-3 (Supplementary Fig 2b) (we were unable to establish a stable cell line expressing the PDGFRA WT in Ba/F3 which was likely due to the low transformation activity). Moreover, stable expression of Y288C in MCF10A markedly increased the size of spheroids in 3D culture without EGF/insulin supplement similar to the known intracellular activating mutants V561D and D842V (Fig. 1e). P345S increased the size of spheroids albeit more modestly. Expression of the activating mutants, especially P345S and to a lesser degree Y288C, significantly enhanced cell migration when compared to MCF10A parental and PDGFRA WT expressing cells (Fig. 1f). The phenotypic changes induced by

each of the activating mutants have been verified with multiple independent stable cell lines. Thus, extracellular domain mutations in PDGFRA have the potential to alter both proliferation and migration, key components of tumor pathophysiology.

**Y288C protein has altered glycosylation and localization.**
Protein glycosylation in the ER and Golgi apparatus (Golgi) is critical for proper protein folding, sorting and trafficking to correct destinations, such as cell surface expression for receptors[32]. Relative N-glycosylation of PDGFRA can be detected as two major bands of different sizes by western blot. The receptor with larger apparent size (upper band) is modified by mature complex glycosylation while the one with smaller apparent size (lower band) is partially glycosylated by mannose[33,34]. As shown in Fig. 1c and Supplementary Fig 2a, unlike PDGFRA WT and other PDGFRA mutants that had comparable expression of both the upper and lower bands, Y288C protein was predominantly located in the lower band in both cell line models. The change in expression pattern was observed in independent stable cell lines expressing Y288C. To further investigate glycosylation of Y288C, protein lysates of MCF10A stable cell lines were treated with glycosidases, endoglycosidase H (Endo H) that cleaves the chitobiose core of high mannose and some hybrid oligosaccharide but not complex glycans and peptide-N-glycosidase F (PNGase F) that cleaves and completely removes all N-linked glycosylation. After Endo H treatment, the lower band, which was the dominant form of Y288C, shifted and the apparent protein size was reduced, consistent with Y288C being primarily high mannose glycosylated (Fig. 2a). After the PNGase F treatment, both upper and lower bands shifted, showing that a small subset of Y288C protein was modified by complex glycosylation (Fig. 2a). As Y288C mutation affected maturation of glycosylation from high mannose to complex glycosylation that contributes to cell surface expression of PDGFRA protein[34], we next examined the effect of the Y288C mutation on PDGFRA cell surface expression. Although altered glycosylation on Y288C did not abolished cell surface expression (Fig. 2b), the amount of Y288C protein expressed on the cell surface was not concordant with total protein expression levels. Indeed, while total Y288C protein levels were much higher than that of WT in MCF10A stable cell lines, cell surface expression was relatively lower. ER and Golgi are the major subcellular compartments where protein glycosylation takes place; therefore, we hypothesized that Y288C protein, which was aberrantly glycosylated, might be retained in the ER or Golgi. Immunofluorescence staining showed that PDGFRA WT was primarily membrane localized with a smaller portion co-localizing with GM130, a marker of Golgi (Fig. 2c). Colocalization with the Golgi was also observed in cells expressing V561D and D842V, as previously reported[35]. On the other hand, Y288C protein had little cell surface expression with a dominant perinuclear localization that co-localized with GM130 as well as

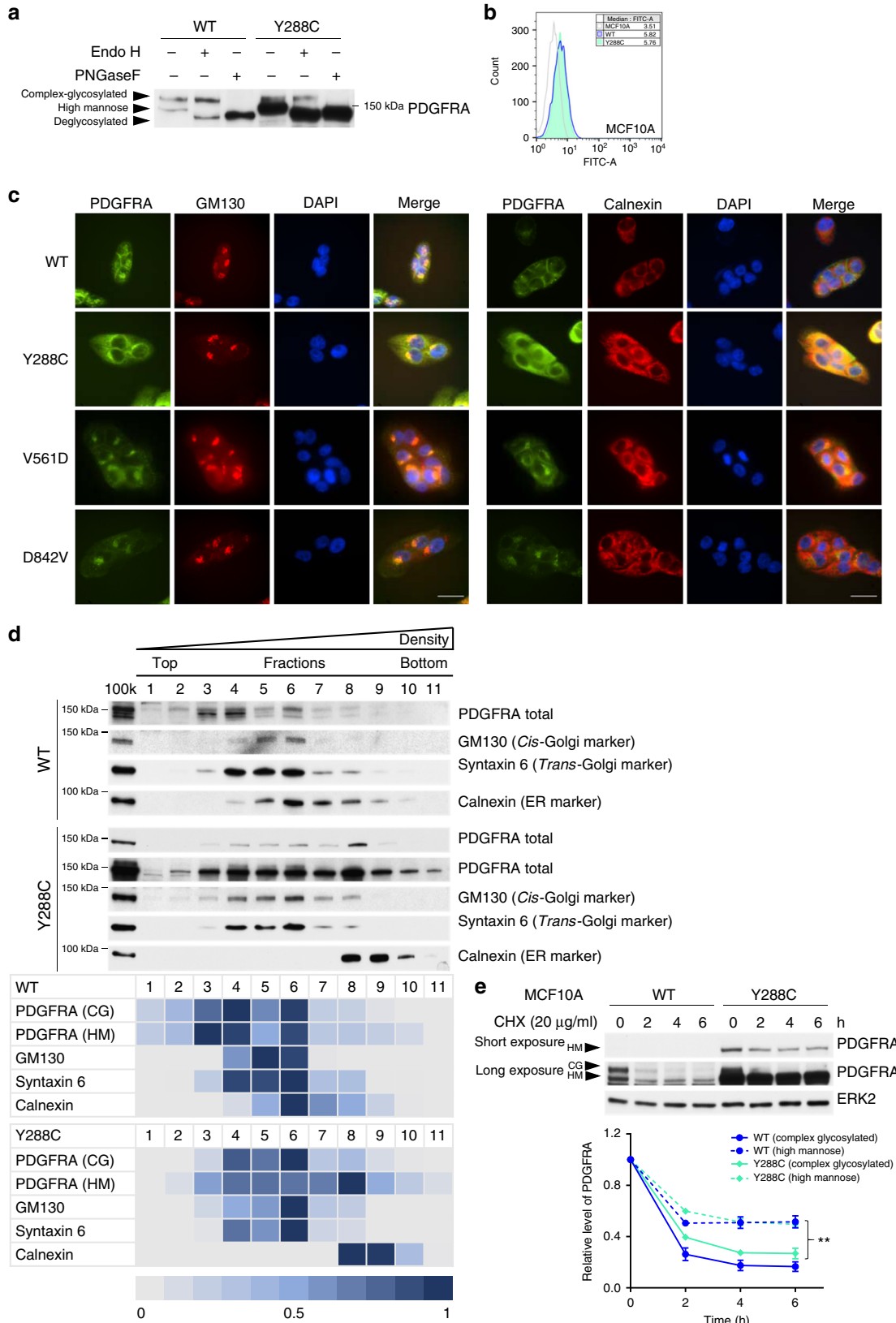

calnexin, an ER marker (Fig. 2c). Localization of PDGFRA to the ER was not detected in cells expressing PDGFRA WT, V561D, or D842V (Fig. 2c). Thus most of the Y288C PDGFRA appears to be trapped in the ER. To further assess localization, we used Opti-Prep gradient ultracentrifugation to separate membrane compartments. Both the cis- and trans-Golgi marker, GM130 and Syntaxin 6, were primarily detected in fractions 4–6 in both PDGFRA WT and Y288C expressing cells. Surprisingly, calnexin, which was mainly detected in fractions 6–8 in MCF10A parental cells (Supplementary Fig 3) or cells expressing PDGFRA WT,

**Fig. 2** Y288C protein has altered glycosylation and localizes at endoplasmic reticulum (ER). **a** Protein lysate of MCF10A expressing PDGFRA WT or Y288C were digested with or without Endo H or PNGase F and the band shift of PDGFRA was analyzed by SDS-PAGE. **b** Cell surface expression of PDGFRA in MCF10A parental (gray), PDGFRA WT (blue) or Y288C (green) expressing cells was assessed by flow cytometry. **c** MCF10A cells expressing PDGFRA WT, Y288C, V561D, or D842V were fixed and stained with anti-PDGFRA (green), anti-GM130 (Golgi marker; red), or anti-calnexin (ER marker; red), and DAPI (nucleus; blue) to visualize the subcellular colocalization of PDGFRA with Golgi and ER. Bar, 25 μm. **d** An OptiPrep gradient ultracentrifugation was used to resolve the membrane fractions (100k) of MCF10A cells stably expressing PDGFRA WT (upper panel) or Y288C (lower panel). 11 fractions were collected. Equal volume of lysate of each fraction were separated by SDS-PAGE and immunoblotted with antibodies against PDGFRA, calnexin (ER marker), GM130 (cis-Golgi marker), and syntaxin 6 (trans-Golgi marker). Heat maps showing the distribution and the relative levels of indicated proteins across different fractions were generated. The fraction with the highest abundance of indicated protein was defined as 1. CG, complex glycosylated; HM, high mannose. **e** MCF10A cells stably expressing PDGFRA WT or Y288C were treated with or without cycloheximide (CHX; 20 μg ml$^{-1}$) for 2–6 h. The level of PDGFRA was analyzed by western blot and ERK2 was used as loading control. The band intensities of complex (solid line) or high mannose (dash line) glycosylated PDGFRA WT (blue) and Y288C (green) in three independent experiments were quantified with densitometry, normalized against ERK2, and presented as mean ± SD. Two-way ANOVA; **$p < 0.01$

shifted to fractions 8–10 in Y288C expressing cells (Fig. 2d), suggesting that Y288C expression increased the density of the ER. Complex and high mannose glycosylated PDGFRA WT were predominantly detected in low-density fractions but not in ER dominant fractions (fraction 7 and 8). In contrast, complex glycosylated Y288C protein was primarily localized in the Golgi enriched fractions (fractions 4–6), while high mannose glycosylated Y288C protein was distributed across the Golgi and ER-enriched fractions but was highly enriched in the high-density ER dominant fraction (Fig. 2d).

As indicated above, Y288C protein was present at higher levels than the WT receptor. The difference in protein amount was independent of PDGFRA transcript abundance (Supplementary Fig 4a) and the amount of virus used to infect the cells (Supplementary Fig 4b). Further, in single cell clones isolated from the stable cell lines established with $1 \times 10^6$ IFU ml$^{-1}$ of virus, transcript levels varied among different single cell clones (Supplementary Fig 4d); however, the amounts of Y288C protein were similar in all clones but higher than WT and dominantly high mannose glycosylated (Supplementary Fig 4c). This further supports the contention that the high mannose glycosylation and increased protein level of Y288C was independent of overexpression.

The retention of Y288C protein in the ER could allow escape from ligand-stimulated degradation[36] resulting in increased protein stability and abundance of the high mannose glycosylated receptor. To test this hypothesis, MCF10A cells expressing PDGFRA WT or Y288C were treated with the protein synthesis inhibitor cycloheximide (CHX) and stability of PDGFRA was assessed. We found that the high mannose glycosylated PDGFRA, either from WT or Y288C, was more stable than the complex glycosylated receptor (Fig. 2e). Thus, Y288C mutation is primarily high mannose glycosylated, which is associated with subcellular retention in the ER and consequent increased protein stability.

**Y288C protein is constitutively active**. Y288 is in the ligand binding Ig-like domain 3. To determine whether Y288C, which was dominantly localized to the ER, was ligand-sensitive, we starved cells and then added PDGF-BB and probed for phosphorylation of PDGFRA and downstream signaling proteins[37]. Parental MCF10A cells have very low abundance of PDGFRA (Fig. 1b) that was insufficient to mediate detectable responses to PDGF-BB (Supplementary Fig 5a). Thus, any changes in signaling pathways induced by PDGF-BB in PDGFRA WT or Y288C expressing MCF10A stable cell lines would be induced by the exogenous expressed receptor.

Phosphorylation at multiple sites in Y288C in MCF10A were maintained in the absence of serum, which contains multiple growth factors including PDGF-ligands, and indeed were not altered by the addition of PDGF-BB. In contrast, phosphorylation of multiple sites in PDGFRA in WT expressing MCF10A cells was low in the presence of serum and for most sites decreased further by serum starvation. In MCF10A expressing PDGFRA WT, there was a modest transient increase in PDGFRA phosphorylation in response to PDGF-BB peaking at 15 min for most phosphorylation sites with the exception of Y742, which was maintained at stable level. Interestingly, phosphorylation of PDGFRA WT was only detected in the complex glycosylated receptor while both complex and high mannose Y288C were phosphorylated with Y720, Y754, and Y1018 mainly presented in high mannose glycosylated Y288C and Y572/574 and Y742 mainly detected in complex glycosylated Y288C. Results with Ba/F3 stably expressing Y288C were similar to those of MCF10A except for a modest decrease in phosphorylation of some PDGFRA tyrosines on serum starvation (Supplementary Fig 5b). Importantly however, in both cell models, Y288C phosphorylation was constitutively elevated and not altered significantly by either serum starvation or addition of PDGF-BB.

Under basal conditions, p-STAT3, p-STAT5, p-Akt, p-ERK1/2, and p-p38 MAPK were markedly higher in cells expressing Y288C than in cells expressing PDGFRA WT (Fig. 3a). Following stimulation with PDGF-BB, phosphorylation levels were transiently increased peaking at 15 min in WT receptor expressing MCF10A cells. In Y288C expressing MCF10A and Ba/F3 cells, p-STAT3, p-STAT5, p-Akt, p-ERK1/2, and p-p38 MAPK were highly phosphorylated in the presence of serum with modest decreases on serum starvation with the exception on p38 MAPK phosphorylation which increased in MCF10A (Fig. 3a) but decreased in Ba/F3 (Supplementary Fig 5b) stable cell lines. PDGF-BB did not alter p-STAT3 or p-STAT5, only modestly increased p-Akt and p-ERK1/2 in both cell line models. Intriguingly, PDGF-BB induced a marked time-dependent decrease in ERK1/2 and p38 MAPK phosphorylation. This is consistent with phosphatases activated by PDGF-BB as part of a feedback loop having access to sites on p-ERK1/2 and p-p38 MAPK in Y288C expressing cells but not having access to PDGFRA[38]. These results suggested that intrinsic constitutive activation rather than extrinsic stimulation with PDGF-ligands was the principal mediator of the effects of Y288C. We thus hypothesized that Y288C would be sufficient to drive cell proliferation and survival in the absence of serum. Indeed, expression of Y288C was able to drive cell proliferation in MCF10A in the absence of serum and exogenous ligands (Fig. 3b). Furthermore, Ba/F3 cells stably expressing Y288C were resistant to apoptosis in serum-free conditions compared to cells expressing other PDGFRA activating mutants (P345S and D842V) as indicated by a lower percentage of DIOC6(3)-low and propidium iodide (PI)-positive cells (Supplementary Fig 5c) and also by lower levels of the apoptotic markers, cleaved caspase-

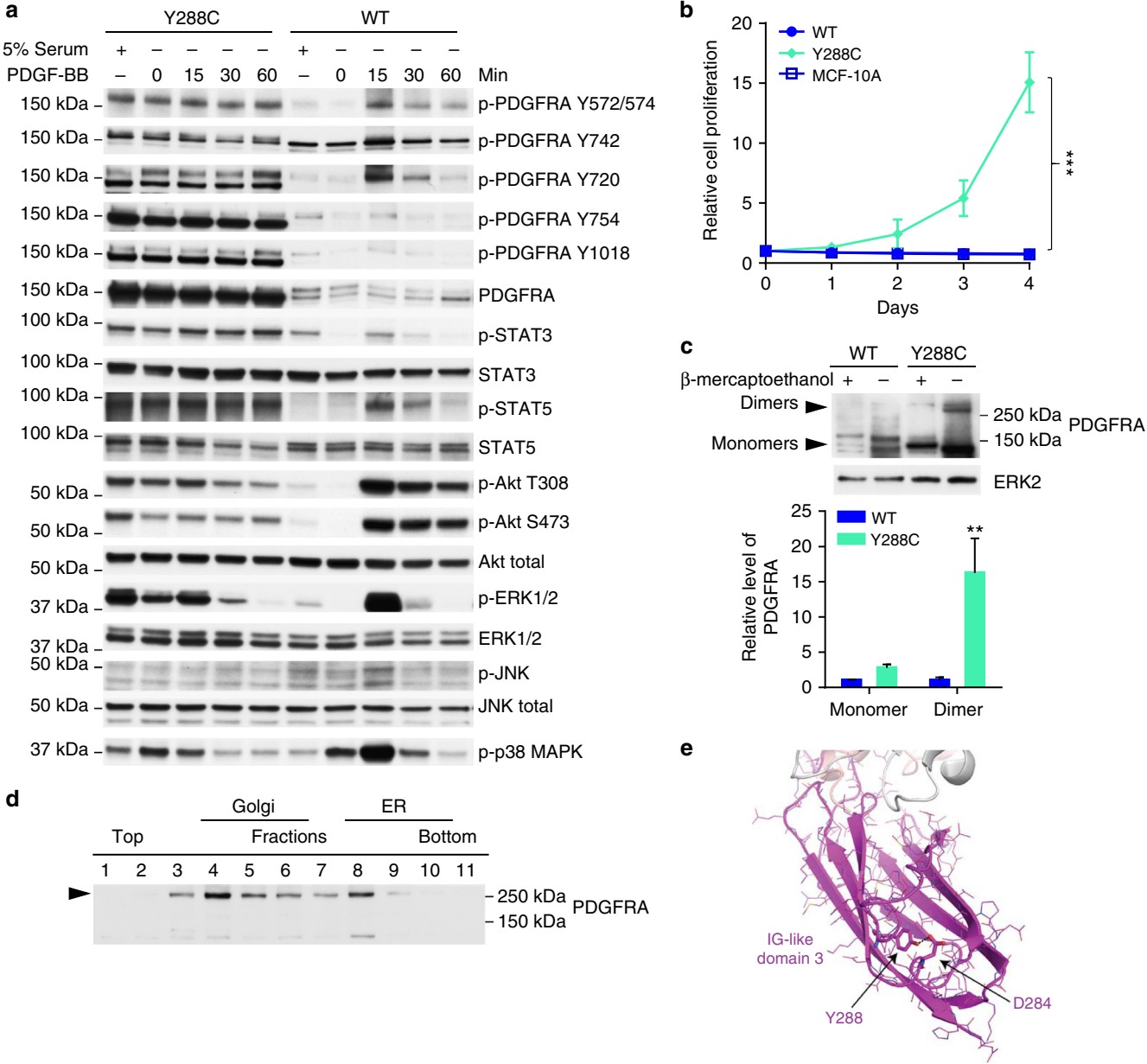

**Fig. 3** Y288C protein is constitutively active at ER and Golgi. **a** MCF10A cells stably expressing PDGFRA WT (right) or Y288C (left) were serum-starved for 16 h and stimulated with PDGF-BB (50 ng ml⁻¹) for 15–60 min. Whole cell lysates were resolved by SDS-PAGE and blotted with indicated antibodies to analyze for the activation of PDGFRA and different downstream signaling pathways. Corresponding total proteins were used as loading controls. **b** Relative cell proliferation of MCF10A parental, PDGFRA WT or Y288C expressing cells in the absence of serum of three independent replicates was assessed for 4 days (exponential growth equation). **c** Protein lysate of MCF10A expressing PDGFRA WT or Y288C were resolved by SDS-PAGE in the presence or absence of β-mercaptoethanol. Monomeric and dimerized PDGFRA were detected by anti-PDGFRA antibody, and ERK2 was used as loading control. The band intensities of monomers and dimers from three independent experiments were measured by densitometry and normalized to ERK2. The levels of PDGFRA monomers and dimers were presented as mean ± SD relative to WT. The statistical significance of dimer vs the corresponding monomer was assessed by Student's $t$-test. **d** Y288C receptor dimerization in ER and Golgi was analyzed by western blot of the OptiPrep gradient ultracentrifugation resolved membrane fractions in the absence of reducing agent. **e** Homology model of the Ig-like domain 3 of PDGFRA predicted a hydrogen bond formed between the carboxylate group of D284 and the hydroxyl group of Y288. **$p < 0.01$; ***$p < 0.001$

7 and cleaved PARP (Supplementary Fig 5d). Together, these data signify the importance of the intrinsic constitutive activation of Y288C in driving cell proliferation and viability.

To investigate the origin of the constitutive activity of Y288C, we examined the formation of PDGFRA dimers, which are required for receptor activation, by western blotting in the absence of reducing agent. As shown in Fig. 3c, Y288C dimers were much more abundant in lysate in the absence of reducing agent when compared to PDGFRA WT. The dimerized Y288C

protein was largely decreased by the addition of reducing agent, which cleaves disulfide bonds between cysteines, suggesting that a disulfide bond between the Y288C dimers or an intramolecular disulfide bond contributes to the dimerization. Strikingly, dimerized PDGFRA Y288C was detected primarily in OptiPrep gradient ultracentrifugation fractions enriched with Golgi and ER (Fig. 3d), suggesting that dimerization occurs primarily in these compartments and that the small amount of PDGFRA that escapes the ER and Golgi and translocates to the cell surface is

primarily monomeric. Indeed, it is likely this limited amount of monomeric PDGFRA Y288C that localized to the membrane that mediates the modest responses to PDGF-BB in terms of cell signaling (Fig. 3a).

**Structural effect of Y288 mutation**. Receptor dimerization mediated by disulfide bonds between unpaired cysteine has been shown in multiple receptors with missense mutations to cysteine at the extracellular domain[16,39–41]. To explore if the mutation to cysteine and the formation of an interchain disulfide bond were the major cause of receptor dimerization and thus the constitutive activation of Y288C, we analyzed the dimerization of another PDGFRA Y288 mutation, Y288H, which lacks the unpaired cysteine for the potential formation of disulfide bond between receptors, found in the TCGA dataset. Similar to Y288C, protein abundance of Y288H was much higher than PDGFRA WT and predominantly high mannose glycosylated (Supplementary Fig 6a). Moreover, Y288H, similar to Y288C, was also predominantly localized around the nucleus and co-localized with GM130 and calnexin (Supplementary Fig 6b), suggesting that mutation of Y288 was critical for the altered PDGFRA glycosylation and subcellular localization. Importantly Y288H dimers were present in protein lysates, albeit lower than Y288C (Supplementary Fig 6a), suggesting the disulfide bond formed between the unpaired cysteine in Y288C is not the only factor contributing to the dimerization of the mutant receptors. The functional activity of Y288H in transient transduction assays was less than that of Y288C in MCF10A (Supplementary Fig 1a) and marginal in Ba/F3 (Fig. 1a). Hence, stable expression of Y288H in MCF10A promoted cell proliferation in the absence of EGF/ insulin, but significantly slower than that of stable expression of Y288C (Supplementary Fig 6c). Thus, the disulfide bond between the unpaired cysteine produced by Y288C is not required for dimerization of PDGFRA or trapping in the ER or Golgi, but does enhance the activity of the Y288 mutated receptor.

We applied structural modeling to explore mechanisms contributing to the intracellular localization Y288 mutations. Since the available crystal structure of PDGFRA contains only the cytoplasmic domain[42], we built a homology model of PDGFRA Ig-like domain 1 to 3 based on PDGFRB (PDB code: 3MJG; 2.30 Å), which is the highest resolution crystal structure with the best sequence similarity to PDGFRA. From the homology model, Y288 of PDGFRA is part of the hydrophobic core of Ig-like domain 3 of PDGFRA. The hydroxyl group of Y288 is predicted to form a hydrogen bond with the carboxylate group of D284, suggesting that Y288 contributes to stabilization of the hydrophobic core (Fig. 3e). Thus, mutation of tyrosine 288 could perturb folding of the Ig-like domain 3 domain potentially exposing the unpaired cysteine in Y288C protein for inter- or intramolecular disulfide bond formation[16]. Nevertheless, Y288 mutations are sufficient to induce dimerization and aberrant localization indicating that disulfide bond formation at Y288C is not required for this process.

**Y288C activates distinct signaling pathways in ER and Golgi**. Based on the observation of dimerized Y288C protein in the Golgi and ER (Fig. 3d), we further investigated the activity of Y288C protein in those two compartments by assessing phosphorylation of Y288C proteins in OptiPrep gradient fractions. Surprisingly, Y288C protein displayed a differential phosphorylation pattern in those two compartments. p-PDGFRA Y572/574 and Y742 were predominantly detected in the Golgi enriched fractions and p-PDGFRA Y720, Y754, and Y1018 were found in both Golgi and ER-enriched fractions (Fig. 4a). Tyrosine phosphorylation of these sites recruits different linker molecules

resulting in distinct downstream signaling effects[43]. Y572/574 are docking sites for STATs and Src family members; Y742 interacts with the p85 regulatory subunit of PI3K and activates PI3K signaling; Y720 and Y754 recruit SHP-2 and thereby activate the MAPK signaling cascade[44]; Y1018 associates with PLC-γ. We thus speculated that the differential phosphorylation pattern of Y288C protein might lead to differential activation of downstream signaling in distinct subcellular compartments. To test this hypothesis, we treated the MCF10A cells expressing PDGFRA WT or Y288C with monensin, which inhibits protein release from Golgi by blocking medial- to trans-Golgi transition, and brefeldin A (BFA), which impedes protein export from the ER. Membrane localization of PDGFRA WT was decreased and both PDGFRA WT and Y288C protein accumulated in Golgi after monensin treatment (Fig. 4b, upper panel). Further, BFA treatment of PDGFRA WT recapitulated the perinuclear location of Y288C protein in untreated cells and increased the colocalization of both PDGFRA WT and Y288C with calnexin (Fig. 4b, lower panel). These results are consistent with the expected effects of monensin and BFA. Phosphorylation of Y288C protein and activation of downstream signaling pathways after monensin and BFA treatments were evaluated by western blotting (Fig. 4c). Monensin caused a band shift and significantly increased phosphorylation of Y288C protein at Y572/574 and its downstream effectors STAT3/5. These effects were completely abolished by BFA, suggesting that enhanced Y288C phosphorylation at Y572/574 in the Golgi, but not in ER activated STAT signaling. Our finding was in line with a previous study which showed that Golgi localized V561D and D842V led to activation of STAT signaling[35]. Y742 phosphorylation demonstrated similar changes to Y572/574 after treatment with monensin or BFA; however, Akt phosphorylation was decreased by monensin and totally eliminated by BFA. This is consistent with Y288C phosphorylation at Y742 on the cell surface or in the Golgi, but not in ER, being able to activate Akt signaling. Monensin and early time points after BFA treatment did not alter size or amount of p-PDGFRA Y720 and Y754. This is consistent with phosphorylation of Y288C at Y720 and Y754 occurring mainly on the high mannose glycosylated receptor in the ER. Inhibition of cell surface PDGFRA localization by monensin or BFA both markedly decreased but did not abolish ERK1/2 phosphorylation, indicating ERK1/2 could be activated by cell surface PDGFRA as well as by PDGFR in the Golgi or ER (Fig. 4c). Our results revealed the differential phosphorylation of Y288C protein and subsequent activation of a complex signaling network in distinct subcellular compartment which was absent in PDGFRA WT expressing cells.

We next sought to determine which downstream signaling pathways, STATs, Akt, or ERK1/2, were required for proliferation induced by constitutively activated Y288C. Knockdown of STAT3, Akt, or ERK1/2 siRNA in MCF10A cells diminished cell viability (Fig. 5a) and suppressed cell proliferation in Matrigel (Fig. 5b). The knockdown efficiency by different siRNAs was verified by Western blot (Fig. 5c). The effects of STAT5 siRNA on STAT5 levels were modest potentially explaining the modest effects of STAT5 siRNA on cell function. Nevertheless, coordinate activation of STAT3, Akt, and ERK1/2 is required for optimal cell proliferation induced by constitutively active PDGFRA.

**PDGFR inhibitors have limited activity against Y288C**. Imatinib and sunitinib are the most common clinically used small molecule inhibitors against mutant *PDGFRA*, with imatinib being the first line therapy and sunitinib being used in patients with imatinib-resistance[45]. To test whether PDGFR inhibitors effectively inhibit Y288C-induced viability or proliferation, MCF10A cells stably expressing Y288C, V561D, and D842V were treated

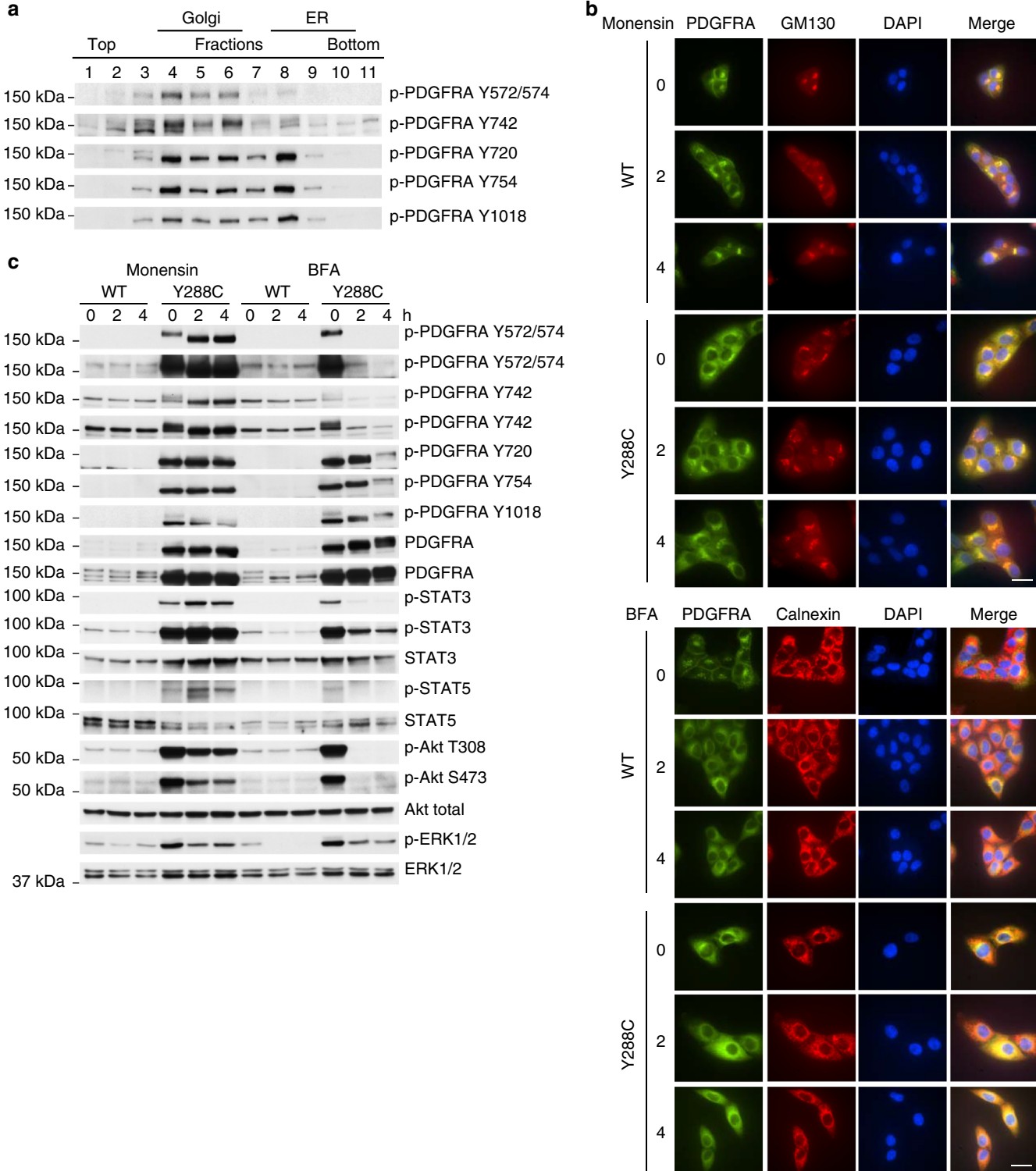

**Fig. 4** Y288C protein activates signaling pathways in distinct subcellular compartments. **a** Phosphorylation of the Y288C receptor at the ER and Golgi was analyzed by western blot of the OptiPrep gradient ultracentrifugation resolved membrane fractions. MCF10A cells stably expressing PDGFRA WT or Y288C were serum-starved for 16 h and treated with or without monensin (500 nM) or BFA (1 μM) for 2 or 4 h respectively. **b** Cells were fixed and stained with anti-PDGFRA (green), anti-GM130 (Golgi marker; red) or anti-calnexin (ER marker; red), and DAPI (nucleus; blue) to examine the subcellular localization of PDGFRA after treatments. Bar, 25 μm. **c** Protein was extracted and analyzed for the activities of Akt, STAT3, STAT5, ERK1/2 signaling pathways by western blot with indicated antibodies. Corresponding total proteins were used as loading controls

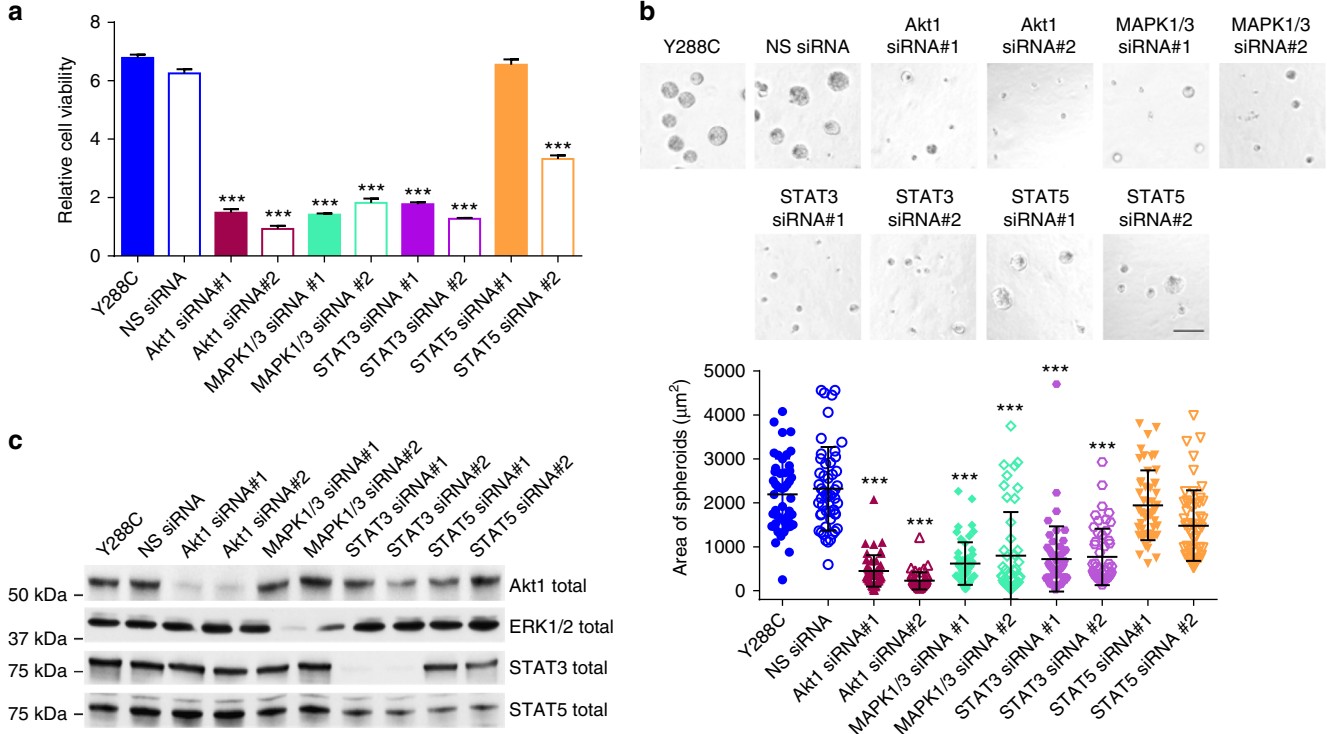

**Fig. 5** Y288C promotes cell proliferation through Akt, ERK1/2, and STAT3. MCF10A cells stably expressing Y288C was transfected with siRNA specifically target, *Akt1*, *MAPK1/3*, *STAT3*, or *STAT5*, respectively. **a** The knockdown efficiency was verified by western blot 48 h after transfection. 24 h after transfection, cells were trypsinized, counted and plated for **b** cell viability assay (Student's *t*-test) and **c** 3D growth in growth factor-reduced Matrigel for 4 days. The relative cell viability of three independent replicates was assessed with Prestoblue 4 days after transfection. Bar, 100 μm. The area of 50 spheroids are presented as mean ± SD (Dunnett's multiple comparisons test). \*\*\**p* < 0.001

with imatinib, sunitinib, and two other PDGFR inhibitors, nilotinib and crenolanib. V561D, an imatinib-sensitive PDGFRA mutant, and D842V, which is resistant to imatinib, but sensitive to crenolanib, were included as controls[9,20,46]. As expected, V561D was sensitive to all PDGFR inhibitors and D842V was resistant to all PDGFR inhibitors except crenolanib (Fig. 6a). Strikingly, as compared to the V561D, Y288C expressing cells were resistant to each of the PDGFR inhibitors, especially Nilotinib. Y288C expressing cells were not as resistant as D842V to imatinib or sunitinib but were remarkably resistant to crenolanib compared to D842V. Sunitinib was the most effective inhibitor for Y288C expressing cells; however, the $IC_{50}$ was 15-fold higher than that for V561D expressing cells (Fig. 6a). The limited efficacies of PDGFR inhibitors against Y288C were also observed in independent MCF10A stable cell lines and Ba/F3 stable cell lines (Supplementary Fig 7). Thus Y288C expressing cells are likely to be resistant to clinically relevant doses of the inhibitors. There is thus a need for alternative approaches to management of tumors expressing PDGFRA Y288C. Since knockdown of Akt, ERK1/2, and STAT3 significantly inhibited the proliferation of cells expressing Y288C (Fig. 5), we assessed whether clinical inhibitors against the Akt, ERK1/2, or STAT3 signaling pathways would be an effective therapeutic strategy against Y288C. All three PDGFRA mutants tested including Y288C were highly sensitive to GSK2126458 (omipalisib), a PI3K/mTOR inhibitor, with $IC_{50}$ in the nanomolar range, suggesting the PI3K/Akt pathway was required for cell viability driven by all three mutations (Fig. 6b). The $IC_{50}$ of GSK1120212B (trametinib), an effective MEK1/2 inhibitor, in cells expressing Y288C was at least 20-fold lower than in cells expressing V561D or D842V, indicating that the Y288C expressing cells were also dependent on MEK/ERK signaling for survival (Fig. 6b). To inhibit STAT3 signaling,

AZD9150, a novel antisense oligonucleotide (ASO) against STAT3 was used[47]. Although the treatment with STAT3 ASO effectively decreased STAT3 protein, the viability of cells expressing Y288C was not affected even at the highest dose of 20 μM while that of cells expressing V561D or D842V were modestly decreased by 22 and 18%, respectively (Fig. 6b). The results suggest that targeting the PI3K or MEK/ERK pathway may provide advantageous for patients with Y288C mutations given the marked resistance to conventional PDGFRA inhibitors.

## Discussion

Over 50% of *PDGFRA* aberrations locate in the extracellular domains in several types of tumors, including glioma and colorectal cancer, suggesting that mutations in the extracellular domain, not only aberrations in the kinase and juxtamembrane domain, are potentially driver aberrations that warrant implementation of targeted therapy. Despite the potential importance of extracellular domain mutations, functional characterization of these *PDGFRA* extracellular domain missense mutations has been unfortunately heretofore lacking. In this study, we assessed the transformation ability of a series of *PDGFRA* missense mutations, including 10 located in the extracellular domain. Strikingly three of these mutations were activating including Y288C and P345S in the extracellular Ig-like domains and P577T in the juxtamembrane domain. Further, we revealed neomorphic effects of Y288C protein including constitutive dimerization, ligand-independent receptor phosphorylation and differential activation of downstream signaling pathways at the ER and Golgi. Strikingly, cells expressing the Y288C mutant are remarkably resistant to all approved PDGFRA inhibitors. Molecular characterization of Y288C signaling provided a potential therapeutic strategy to target the Y288C mutation.

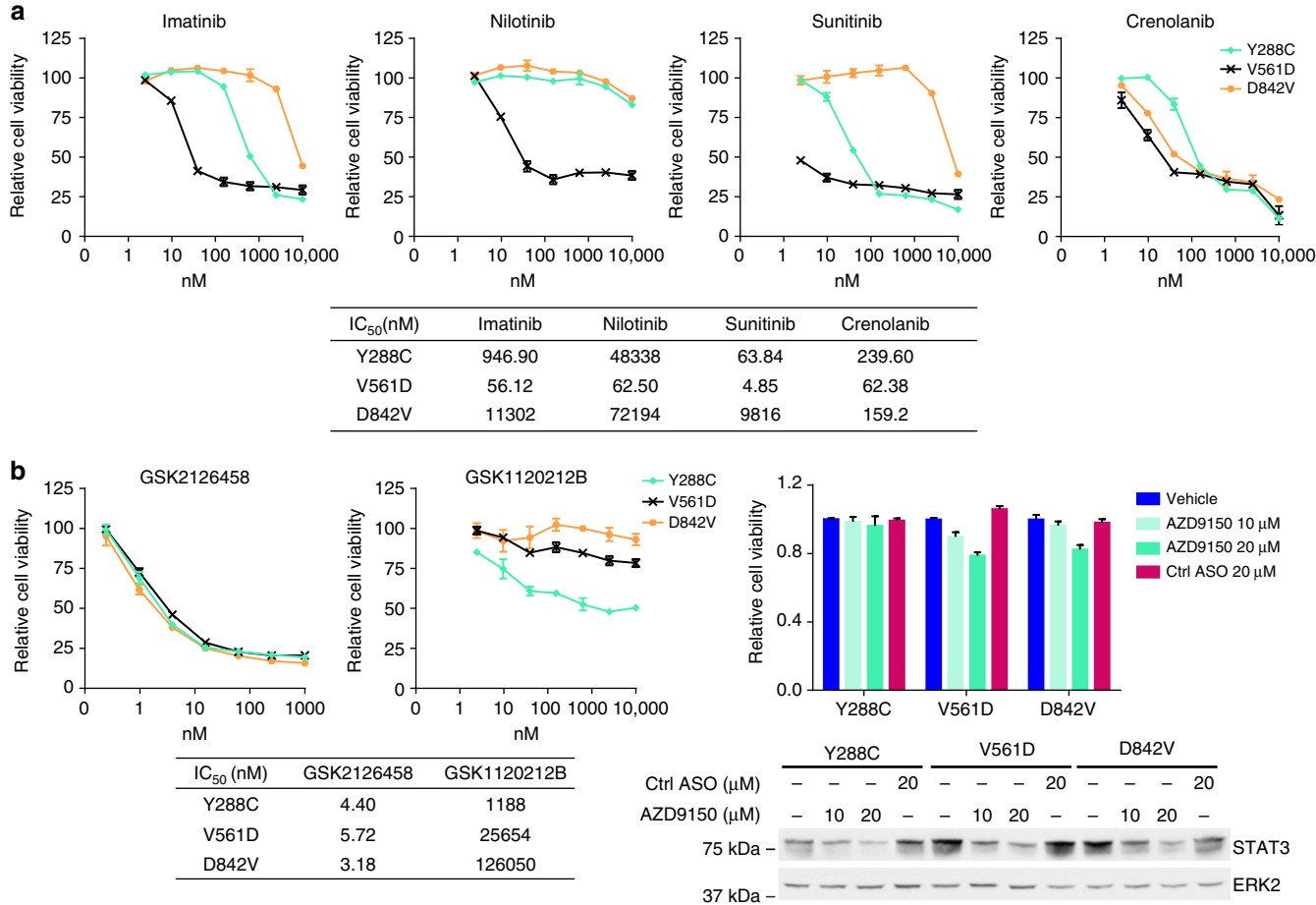

**Fig. 6** PDGFR inhibitors have limited efficacy against Y288C mutation. MCF10A cells expressing PDGFRA mutants were treated with DMSO or 2.44nM–10 μM of **a** PDGFR inhibitors (Imatinib, Nilotinib, Sunitinib, and Crenolanib), **b** PI3K/mTOR inhibitor (GSK2126458), MEK inhibitor (GSK1120212B), or STAT3 ASO (AZD9150) for 3 days. Each treatment was performed in triplicate and cell viability is presented as mean ± SD. The IC50 was calculated with Graphpad Prism 6

Altered glycosylation together with increased protein stability, localization at the ER and Golgi and constitutive dimerization and activation represent the major characteristics of Y288 mutated PDGFRA. Unlike another *PDGFRA* extracellular domain mutation, W349C, which totally abolishes complex glycosylation and cell surface expression[12], Y288 mutated PDGFRA is predominantly incompletely glycosylated with a small component undergoing complex glycosylation and is able to escape the ER and Golgi and localize to the membrane. Incomplete glycosylation and gain of localization at intracellular membranes as a consequence of mutations have been reported in multiple receptor tyrosine kinases, such as anaplastic lymphoma kinase[48], c-Kit[49,50], CSF1R[13] and Fms-like tyrosine kinase 3 (FLT-3)[51]. Inefficient protein folding and constitutive phosphorylation of the mutated receptor during biosynthesis have been posited to explain trapping in the ER as well as incomplete glycosylation of different tyrosine kinase receptors[13,48,50,51]. Based on homology modeling that suggests Y288 might stabilize the hydrophobic core of the Ig-like domain 3, mutation of Y288 might decrease the efficiency of protein folding leading to enhanced interaction of the mutated receptor with ER-resident chaperones preventing degradation of the mutated receptor while concurrently resulting in ER retention[51,52].

Constitutive phosphorylation of mutated FLT-3 at the ER impedes maturation of glycosylation during receptor biosynthesis[51]. Overexpression of ER-resident phosphatases, PTP1B, SHP-1, and PTP-PEST, or inhibition of receptor phosphorylation with

tyrosine kinase inhibitors increased complex glycosylation of the mutant FLT-3[51]. Thus, constitutive dimerization and subsequent autophosphorylation at Y720, Y754, and Y1018 of PDGFRA Y288 mutations at the ER could contribute to the limited complex glycosylation of PDGFRA Y288 mutations.

PDGFRA Y288C retained in the Golgi acquires phosphorylation at Y572/574 and Y742. Importantly, differential phosphorylation of any constitutively activated tyrosine kinase receptor in different subcellular compartments has not been previously demonstrated consistent with this representing a new paradigm. PDGFRB Y751, corresponding to PDGFRA Y742, is hyperphosphorylated in PTP1B knockout cells[53]. It is possible that PTP1B is functional in the ER but not in the Golgi contributing to the differential phosphorylation. Further, the corresponding downstream linker molecules, including Src[54], PI3K[55,56], and Ras[57,58] in the Golgi and the ER, could associate with the differentially autophosphorylated PDGFRA mutant and subsequently activate STAT, Akt, and ERK signaling in different subcellular compartments. The differential autophosphorylation elucidated in this study provides a molecular mechanism to explain the activation of alternate signaling pathways in cells expressing mutated constitutively active receptor tyrosine kinases that are trapped at intracellular membranes[16,50,56].

It is important to note that Y288C is sufficient to induce cell proliferation and survival in the absence of serum or exogenous growth factors suggesting that constitutive dimerization and

activation of the ER and Golgi trapped receptor may be sufficient for cell proliferation. Cooperation of the intrinsic signaling with the extrinsic activation of Akt and ERK1/2 signaling through Y288C on the cell surface could drive rapid cell proliferation and potentially contribute to poor outcomes in patients including high-grade glioma where Y288C has been recurrently observed[11,31]. Targeted therapies against PDGFRA are in clinical trials for glioma and approved for metastatic GIST. Here, we provided evidence that Y288C is resistant to PDGFR inhibitors suggesting that the selection of inhibitors for targeted therapy should be conditioned on the specific mutation present rather than globally on the presence of mutations in a gene. Based on our deep dissection of molecular mechanisms underlying function of Y288C, targeting the ERK signaling activated by both extrinsic and intrinsic signaling from Y288C with MEK inhibitors instead of PDGFR inhibitors may be a more specific and effective approach to manage tumors expressing Y288C mutations. This study, together with others[59–61], provides a cautionary tale that targeted therapy may not benefit patients if the treatment decision is only based on the presence of a mutated gene but not the individual mutation. Indeed, neomorphic mutations are likely to require deep mechanistic exploration to identify appropriate therapeutic options.

Our study demonstrates an extracellular domain mutation of a cell surface receptor tyrosine kinase which displays a different molecular mechanism and thus therapeutic sensitivity from those of juxtamembrane and kinase domain mutations, highlighting the importance of characterization of individual mutations in pursuing optimum personalized cancer therapy.

## Methods

**Cell culture and reagents.** Ba/F3 parental (M.D. Anderson Characterized Cell Line Core facility) cells were cultured with RPMI1640 supplemented with 5% fetal bovine serum (FBS) and 5 ng ml$^{-1}$ mouse IL-3. MCF10A parental cells (ATCC® CRL-10317™) were cultured at 37 °C in DMEM/F12 medium supplemented with 0.5 mg ml$^{-1}$ hydrocortisone, 100 ng ml$^{-1}$ cholera toxin, 20 ng ml$^{-1}$ EGF, 10 µg ml$^{-1}$ insulin, and 5% horse serum (growth medium). Ba/F3 parental cells were validated based on continued dependence on IL-3 for propagation and MCF10A parental cells were authenticated by Short Tandem Repeat (STR) analysis at M.D. Anderson Characterized Cell Line Core facility. All cell lines used were tested and free of mycoplasma contamination. The transformation activity of PDGFRA WT or mutants was studied using MCF10A and Ba/F3 cells as described previously[19]. In brief, MCF10A and Ba/F3 cells were spinoculated with lentivirus carrying PDGFRA WT or mutant generated by LentiX-293T cells. After spinoculation, MCF10A cells were cultured in MEBM basal medium (Lonza #CC-3151) with 100 ng ml$^{-1}$ cholera toxin and 52 ng ml$^{-1}$ bovine pituitary extract (Lonza #CC-4009) while Ba/F3 cells were resuspended in RPMI1640 supplemented with 5% FBS without IL-3. Cell viability was measured 7 days after infection and the transforming activity of mutants was compared to viability of cells infected with lentivirus carrying PDGFRA WT. To generate stable cell lines expressing PDGFRA WT or mutants, infected MCF10A cells were selected with Puromycin (2.5 µg ml$^{-1}$) for 7 days and maintained in growth medium. Infected Ba/F3 cells that survived in the transient transduction assay 7–10 days after infection were collected as stable cell lines and cultured in 5% serum medium without IL-3. Experiments with MCF10A parental and stable cell lines were adapted overnight and carried out in assay medium (growth medium without EGF/insulin). Experiments using Ba/F3 cells stably expressing PDGFRA mutants were carried out in 5% serum medium without IL-3. The expression of PDGFRA WT or mutants in stable cell lines was verified with western blot and Real-time PCR. All the stable cell lines were sequenced and confirmed to express only the mutation of interest. To study the effect of ligand stimulation, cells were starved overnight with serum-free medium and stimulated with 50 ng ml$^{-1}$ PDGF-BB (PeproTech) for 15–60 min. The stability of PDGFRA WT and Y288C was assessed with cycloheximide (CHX; 20 µg ml$^{-1}$) treatment for 2–6 hr. Monensin (Sigma) and Brefeldin A (Thermo Fisher Scientific) were used to study the signaling activated at the Golgi and ER.

**Distribution and feature prediction of *PDGFRA* mutations.** Data on *PDGFRA* mutations in the AACR Project Genomics Evidence Neoplasia Information Exchange (GENIE) across different tumor types were downloaded on July 2017 via the Synapse portal. The topology of the mutated amino acid was determined based on the definition in Uniprot KB (P16234). Rank score of mutation features were calculated by ANNOVAR[62] software. Boxplot for activating mutations versus non-activating mutations were drawn with R package "ggpubr". The classification of mutations on 3D structural were predicted by dSysMap[18].

**Cell proliferation/ viability assay.** MCF10A (2500 cells per 100 µl per 96-well) or Ba/F3 ($1 \times 10^4$ cells per 1.5 ml) parental of PDGFRA WT or mutant stable expressing cells were cultured in triplicate with 5% serum medium without supplement of EGF/insulin for MCF10A or IL-3 for Ba/F3 cells. To evaluate MCF10A cell proliferation in the absence of serum, culture medium was removed after overnight incubation, washed twice and replaced with serum-free medium without EGF/insulin. For Ba/F3 cells, 100 µl of cells of each replicate were transferred to 96-well plate before cell viability assay. Cell proliferation/ viability was evaluated with IncuCyte ZOOM® imaging or PrestoBlue (Thermo Fisher Scientific) every 24 h.

**Three-dimensional (3D) cell culture.** $1 \times 10^4$ cells ml$^{-1}$ were resuspended in medium without EGF/insulin and 400 µl of cells were embedded into 8-well chamber-slides coated with growth factor-reduced Matrigel (Corning). Cells were grown for 4–10 days with replacement of medium every 4 days. Five fields of each chamber were captured with EVOS XL Core imaging system equipped with ×10 objective lens. The number of spheroids per field were counted and the size of 50 spheroids was measured using image processing package Fiji.

**Migration assay.** Migration assay was performed using 8 µm pore size 24-well Transwell inserts (Millipore, Bedford, MA). $2 \times 10^5$ MCF10A parental or stably expressing PDGFRA WT or mutant cells were suspended in 400 µl serum-free medium and placed in upper chamber. Medium containing 5% serum without EGF or insulin was placed in the lower chamber as chemoattractant. After 20 h of incubation, cells were fixed and stained with Diff-Quik™ Stain Kit (Simens Healthcare Diagnostic) following manufacturer's instruction. Non-migrated cells on the upper side of membrane were removed with cotton swab. Five fields of the membrane of each replicate were captured and the number of migrated cells per field was counted and presented as mean ± SD.

**Western blotting.** Cells were lysed with SDS sample buffer (65 mM Tris-Cl pH 6.8, 2% SDS, 10% glycerol) supplemented with Halt Protease and Phosphatase Inhibitor Cocktail (Thermo Fisher Scientific). Equal amount of protein were resolved by SDS-PAGE and transferred to PVDF membrane. Membranes were blocked and incubated overnight at 4 °C with the following antibodies: anti-p-PDGFRA (Y572/574; Cat# ab5443; 1:1000), anti-p-PDGFRA (Y720; Cat# ab65258; 1:1000), anti-p-PDGFRA (Y742; Cat# ab5452; 1:1000), anti-calnexin (Cat# ab31290; 1:1000) (Abcam), anti-PDGFRA (Cat# sc-431; clone 951; 1:1000), anti-ERK2 (Cat# sc-154; C-14; 1:2000) (Santa Cruz), anti-p-PDGFRA (Y754; Cat# CST2992; 23B2; 1:1000), anti-p-PDGFRA (Y1018; Cat# CST4547; 1:1000), anti-p-STAT3 (Y705; Cat# CST9131; 1:1000), anti-STAT3 (Cat# CST4904; 1:1000), anti-p-STAT5 (Y694; Cat# CST9351; 1:1000), anti-STAT5 (Cat# CST9363; 1:1000), anti-p-Akt (S308; Cat# CST2965; 1:1000), anti-p-Akt (S473; Cat# CST9271; 1:1000), anti-Akt (Cat# CST4691; 1:1000), anti-Akt1 (Cat# CST2938; 1:1000), anti-p-ERK1/2 (Cat# CST9101; 1:1000), anti-ERK1/2 (Cat# CST4695; 1:1000), anti-p-JNK (Cat# CST4671; 1:1000), anti-JNK (Cat# CST9252; 1:1000), anti-p-p38 MAPK (Cat# CST9211; 1:1000), anti-p38 MAPK (Cat# CST9212; 1:1000), anti-PARP (Cat# CST9542; 1:1000), anti-Caspase 7 (Cat# CST 9492; 1:1000), anti-syntaxin 6 (Cat# CST2869; 1:1000) (Cell Signaling), anti-GM130 (BD Transduction Laboratories; Cat# 610822; 1:500). HRP-conjugated goat anti-mouse (Cat# 1706516; 1:5000) and goat anti-rabbit (Cat# 1706515; 1:5000) antibodies were purchased from Bio-Rad Laboratories. Proteins were visualized by enhanced chemiluminescence (PerkinElmer). Composite figures were prepared with Adobe Photoshop CC. Densitometry was analyzed with Fiji. Uncropped blots are provided in Supplementary Figs. 8–13.

**Immunofluorescence staining and microscopy.** Cells were fixed in 4% paraformaldehyde for 20 min at room temperature followed by permeabilization with 0.1% Triton X-100 for 1 h. Cells were blocked with 5% BSA and incubated with mouse anti-GM130 (1:100) or mouse anti-calnexin (1:200) for 1 h at room temperature and followed by Alexa Fluor 568 donkey anti-mouse (1:400; Molecular Probes) and Alexa Fluor 488 conjugated anti-PDGFRA (D13C6, 1:100; Cell Signaling) after washing with PBS. Stained cells were sealed in mounting medium with DAPI (Vector Laboratories). Images were acquired with Nikon Eclipse TE2000 equipped with 100X oil-immersion objective. Composite figures were prepared with Fiji.

**Cell surface expression.** To evaluate the cell surface expression of PDGFRA in parental MCF10A or Ba/F3 or cells stably expressing PDGFRA WT or Y288C, cells were stained with Alexa Fluor 488 conjugated anti-PDGFRA (1:100; Cell Signaling) and analyzed with flow cytometry (BD FACSAriaII).

**DIOC(6)/propidium iodide staining.** $9 \times 10^5$ cells of Ba/F3 cells stably expressing PDGFRA WT or mutant were cultured in 3 ml of medium with or without 5% serum for 16 h. Cells were collected by centrifugation and stained with 0.1 µM DIOC(6) and 100 µg ml$^{-1}$ PI (Molecular Probes, Life Technologies) at 37 °C for 15 min to determine mitochondrial membrane potential and late apoptosis after serum starvation for 16 h. Cells cultured in medium with 5% serum were served as control. Percentage of cells with decreased membrane potential (DIOC(6)-low) and

late apoptotic cells (PI-positive) were analyzed with flow cytometry (BD FACSAriaII).

**Analysis of protein glycosylation**. MCF10A cells expressing PDGFRA WT or Y288C were lysed with Pierce™ IP Lysis buffer (Thremo Fisher Scientific). Protein deglycosylation with PNGase F or Endo H (New England BioLabs) was carried out according to the manufacturer's instructions. For each reaction, 20 μg of protein was resuspended in lysis buffer supplemented with 1X Glycoprotein Denaturing Buffer and denatured at 100 °C for 10 min. Denatured protein was incubated with or without Endo H or PNGase F at 37 °C for 1 h. The reactions were stopped with Laemmli sample buffer. Deglycosylation of PDGFRA was assessed by mobility shift on SDS-PAGE gel.

**Membrane OptiPrep gradient fractionation**. Five 15-cm dishes of cells ($8 \times 10^6$ cells per dish) were harvested in 2.5 ml of isotonic homogenization buffer (10 mM HEPES, pH 7.8, with 0.25 M sucrose, 1 mM EGTA, and 25 mM potassium chloride; Sigma-Aldrich) supplemented with Halt Protease and Phosphatase Inhibitor Cocktail (Thermo Fisher Scientific) and homogenized with 10 strokes in dounce homogenizer followed by 8 passages through a 25 G needle. Nuclei and unbroken cells were removed by centrifugation at $1500 \times g$ for 10 min at 4 °C. The post-nuclear supernatants were then centrifuged for 1 h at $100,000 \times g$ to collect the membrane fraction. The pellets were resuspended in 1 ml of homogenization buffer and overlaid on the top of the discontinuous OptiPrep gradient prepared by diluting 60% (w/v) OptiPrep Density Gradient Medium (Sigma-Aldrich) with homogenization buffer. The OptiPrep gradient was built as follows: 0.5 ml 30%, 1 ml 25%, 2 ml 20%, 2 ml 15%, 2 ml 12.5%, 1.5 ml 10%, 1 ml 7.5%, 0.5 ml 5%, and 1 ml 2.5%. The OptiPrep gradients with samples were centrifuged at $150,000 \times g$ for 3 h. 11 fractions, 1.1 ml each were collected. The distribution of Golgi, ER, and PDGFRA was assessed by resolving equal volume of samples from each fraction in SDS-PAGE.

**RNA interference**. MCF10A cells stably expressing Y288C were plated into 6-well plate ($2 \times 10^5$ cells per well) overnight before transfection. 20 nM of siGENOME siRNA against human Akt1 (D-003000-05-0002, D-003000-07-0002), MAPK3 (D-003592-01-0002, D-003592-02-0002), MAPK1 (D-003555-01-0002, D-003555-03-0002), STAT3 (D-003544-02-0002, D-003544-03-0002), STAT5A (D-005169-01-0002, D-005169-02-0002) from GE Dharmacon were transfected using RNA IMAX (Thermo Fisher Scientific) according to manufacturer's instruction. siGENOME Non-targeting siRNA (D-001210-05-05; GE Dharmacon) was used as control. 24 h after transfection, cells were trypsinized, counted, and plated for cell proliferation assay and 3D cell culture. Knockdown efficiency was assessed by western blot 48 h after transfection.

**Real-time PCR**. Total RNA was extracted with RNeasy Plus mini kit (Qiagen). 2000 ng of RNA was reverse transcribed to cDNA using the High-Capacity cDNA Reverse Transcription Kit (Thermo Fisher Scientific). Expression of *PDGFRA* and housekeeping gene, *ACTB*, was assessed by TaqMan Real-time PCR assay. Real-time PCR was carried out with TaqMan Fast Universal PCR Master Mix and TaqMan probe against *PDGFRA* (Hs00998018_m1) and *ACTB* (Hs99999903_m1) in MasterCycler RealPlex[4] (Eppendorf). The expression of *PDGFRA* was calculated with the equation $2^{(-\Delta\Delta ct)}$.

**Drug sensitivity assay**. MCF10A cells stably expressing PDGFRA mutants (5000 cells per 100 μl) were seeded in 96-well plates in 5% serum medium without EGF/insulin. Cells were treated with DMSO or inhibitors (2.44 nM to 10 μM) for 72 h. Cell viability was determined using Presto Blue (Invitrogen) at 530 nm excitation/604 nm emission. Cell viability relative to DMSO treated cells was calculated. Each treatment was performed in triplicate. The IC50 was calculated with Graphpad Prism 6. Imatinib, Nilotinib, Sunitinib, Crenolanib, GSK2126458 (PI3K/mTOR inhibitor), and GSK1120212B (MEK inhibitor) were from Selleck. The inhibitors were dissolved in DMSO and stored as 10 mM aliquots at −20 °C. AZD9150 (antisense oligonucleotide against human STAT3) and the control nontargeting antisense oligonucleotide (ASO) were provided by AstraZeneca and Ionis Pharmaceuticals as 4.2 mM and 1.8 mM stock solution, respectively. The ASOs were formulated in PBS and the stock were aliquoted and stored at −20 °C.

**Homology modeling**. The homology model of PDGFRα extracellular region was prepared using MOE v2015.10 software (Chemical Computing Group Inc.) to produce 10 models with medium level refinement using the published structure of PDGFRβ (PDB code 3MJG) as the template structure. The top-scoring model was selected automatically and medium level energy minimization applied to generate the final homology model.

**Statistical analysis**. All experiments were carried out at least three times. Statistical significance was determined using Graphpad Prism 6. Dunnett's multiple comparisons test was used to determine statistical significance of the functional activities of *PDGFRA* mutations in the transient transfection assay, 3D cell culture, and migration assay. Exponential growth equation was used to determine statistical significance of time course of cell proliferation and viability. Two-way ANOVA

was used for statistical significance of the stability between complex glycosylated and high mannose glycosylated receptor. *p*-value in other experiments was calculated using Student's *t*-test. $p < 0.05$ was considered to be statistically significant on the basis of at least three independent sets of experiments.

## Data availability

The data that support the findings of this study are available from the corresponding author upon reasonable request.

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

## Acknowledgements

This work was supported by a training fellowship from the Computational Cancer Biology Training Program of the Gulf Coast Consortia (CPRIT grant number RP170593) to C.K.I., Cancer Prevention and Research Institute of Texas (CPRIT) New Investigator Grant RR160021 and Alfred P. Sloan Research Fellowship FG-2018-10723 to N.S., and NCI grant number U01CA217842-01 and CPRIT grant number RP170640 to G.B.M.

## Author contributions

C.K.I. did project administration, carried out all other experiment and data analysis, and wrote the manuscript. P.K.N, K.J.J, and S.H.S. performed transient transduction assays. Z.J performed analysis on patient sequencing data. P.G.L performed structural analysis. X.H. performed and N.S. supervised the computational analysis on the features of activating mutations. C.p.V. provided support on resources acquirement. R.W. and K.L.S provided critical resources. G.B.M supervised the study, edited the manuscript, and acquired funding.

## Additional information

**Competing interests:** G.B.M. receives sponsored research from AstraZeneca and is on the SAB of AstraZeneca. The remaining authors declare no competing interests.

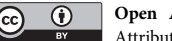

