## [Peer Review File · Nature Communications]

Reviewers' comments:

Reviewer #1 (Remarks to the Author):

In this manuscript, Carman Ip and colleagues have characterized 16 PDGFRA mutations that had been previously identified in various sequencing projects. Three of these mutations confer a gain of function, and one of them, Y288C, accumulates in the ER, leading to constitutive dimerization and signaling, suggesting a new mechanism of oncogenic activation for this receptor. Interestingly, the Y288C mutant confers resistance to tyrosine kinase inhibitors. This study highlights the importance of functionally characterizing mutations that are found by high throughput sequencing efforts.

Comments:

- What criteria were used to select the mutations that were studied?
- The method used to produce infected cells is not available (citation 18 is "in press" and I could not find it online). What was the infection efficiency? It is not clear whether the authors produced several independent cell lines for each of the key mutants.
- In figure 1a and suppl table 1, the location (extracellular/kinase/other) of all reported variants is shown. Since the present study suggests that a large part of these variants may not confer any advantage (likely passenger mutations), it would be interesting to plot the number of confirmed gain-of-function drivers in each category and each tumor type. Also, what is the total number of analyzed samples, including wt, for each tumor type?
- In suppl figure 1, an arbitrary threshold was used. Did the authors perform a statistical analysis to validate the results?
- A striking result of this study is the resistance of the Y288C mutant to TKI (figure 6). I would be important to confirm these key results using Ba/F3 cell lines as an independent model.
- Several studies have reported that kinase transgenes may acquire additional unwanted mutations, particularly when using retroviral vectors (Watanabe-Smith et al, *Oncotarget*, 2017, Vol. 8, pp: 12596-12606). The authors should therefore sequence the transgene in stable cell lines. This is particularly important for the drug sensitivity tests shown in figure 6, because additional mutations may be responsible for the observed resistance to TKI.
- It is difficult to conclude from Suppl fig 7 on the presence or absence of a disulfide bridge to stabilize the Y288C dimer.
- Suppl figure 5 shows constitutive signaling in Ba/F3 cells expressing Y288C. Because the level of residual signaling is highly variable, depending on the starvation conditions, a control cell line is needed (Ba/F3-empty vector and/or WT). It is also unclear whether this "stable" cell line is grown in the absence of IL-3.
- Bahlawane et al published an interesting study on the same topics in *Cell Communication and Signaling* (2015, v13:pp21-), which should be cited. Reference 51 (a conference abstract) has not been confirmed by a full publication.

Reviewer #2 (Remarks to the Author):

This study by Ip and colleagues seeks to clarify the role of PDGFRA in tumor progression. Performing functional characterization of 16 novel PDGFRA mutations (9 extracellular, 2 juxtamembrane, 3 kinase, 2 C-terminal) identified from different tumor types. Compared to established activating mutations (V561D and D842V), 3 mutations (Y288C, P345S, P577T) displayed activity in the IL-3 Ba/F3 model (2 in the EGF/Insulin MCF10A model). Focusing on two extracellular domain mutations (Y288C and P345S), and subsequently Y288C, the authors demonstrate that PDGFRA (Y288C) strongly promotes various cancer hallmarks (eg cell proliferation, viability, spheroid formation). Subsequently analysis revealed that Y288C is largely mannose glycosylated, that may arise as a consequence of a maturation 'block' from high mannose to complex glycosylation. They show that by cell fraction and immunofluorescence that Y288C is retained by the ER/Golgi, which may lead to increased PDGFRA stability (escape from

ligand stimulated degradation), receptor dimerization and hence higher signalling activity in a ligand-independent fashion. This study is timely as it seeks to functionally assess, in a systematic fashion, novel receptor tyrosine kinase (RTK) associated mutations. One notable and very interesting finding is that Y288C proteins in different cellular compartments (Golgi, ER) have different phosphorylation patterns, predicted to result in engagement/activation of distinct effector proteins (of which STAT3, Akt and ERK1/2 appear to be the most important for optimal cell proliferation). Finally, challenging Y288C expressing cells with mutant PDGFRA inhibitors (imatinib, sunitinib) suggests that this mutation causes resistance (albeit to different degrees) to the conventional PDGFRA therapies. In contrast, Y288C mutations were sensitive to 'downstream' therapies, such as PI3K/mTOR, MEK1/2, and STAT3.

Major Comments

- 1) Of the 16 mutations tested, 3 were associated with 'activating' PDGFRA activity. This raises the possibility that the remaining 13 may be bystander or passenger mutations. I would like to see comprehensive analysis, guided by the functional data, to assess if there are any sequence features (eg location, conservation, crystal structure, rate of recurrence, etc) that might have predicted that the 13 are different from the 3. Such information may prove important in improving DNA sequenced based bioinformatics algorithms to discern true drivers from passengers.
- 2) For the MCF10A experiments, it is not clear if the authors tested all 16 mutations, or just the 3 that were positive in the BA/F3 model. If the latter, it would be interesting to test all 16 mutations, as there is a possibility that some of the 'negative 13' may show activity in a different cell type.
- 3) To what extent are the neo-morphic feature of Y288C (including Golgi/ER retention) a consequence of very high levels of stable expression? Conceptually, it remains possible that if wild-type PDGFRA is also expressed to the same amount seen in the Western blots, then this would result in Golgi trapping, signalling etc. This could be addressed either by titrating the amount of Y288C to levels comparable to WT PDGFRA, and then comparing the two; (or in a more challenging fashion, CRISPR engineering PDGFR to harbour the Y288C mutation). This Reviewer's point is that if it is indeed a consequence of 'simple high overexpression', then there is a concern about the in vivo relevance of the study, given that the results are based cell line transfections that usually result in non-physiological levels of overexpression anyway.
- 4) I would like to see the Discussion expanded with more implications for cancer therapy. In an age where there is often a rush to treat patients based on sequence-based mutation data alone, it is conceivable that patients with Y288C mutations might be treated with PDGFRA inhibitors, which by the investigator's data is precisely something *not* to do. Hence, this study is a good cautionary tale about the potential pitfalls of relying on genomic results in isolation.

Minor Comments

- 1) More info could be provided about the 16 mutations tested. Were they selected using some criteria, or do these 16 represent the totality of novel PDGFRA mutations identified in TCGA/GENIE? MDACC? Are these 16 PDGFRA mutations associated with any specific tumor types? Are they recurrent or singletons?
- 2) Data showing the initial selection of the 3 Ba/F3 positive mutations needs to be shown as part of Main Figure 1, and compared to the other 13 mutations.
- 3) Is the following a typographical error? The first part of the sentence seems to contradict the second – "In MCF10A and Ba/F3 cells stably expressing PDGFRA WT (we were unable to establish a stable cell line expressing the PDGFRA WT in Ba/F3 which was likely due to the low transformation activity)". Can Ba/F3 cells be stably transfected or not?
- 4) Is it possible to remine some of the primary tumor data (if it is available) to ask if tumors with

the activating PDGFRA mutations also have high expression of PDGFRA in the same tumor?

Response to referees (NCOMMS-18-01829)

Referring to Reviewer #1's comments:

- What criteria were used to select the mutations that were studied?

We selected mutations in PDGFRA that have not been previously studied or functionally characterized. This has been clarified (p.5, line 4-5).

- The method used to produce infected cells is not available (citation 18 is "in press" and I could not find it online).

Citation 18 (updated as citation 19 in the revised manuscript) is now available online ([http://www.cell.com/cancer-cell/fulltext/S1535-6108\(18\)30021-7](http://www.cell.com/cancer-cell/fulltext/S1535-6108(18)30021-7)). We have now also included the procedure for the transient transduction assay and the generation of the stable cell lines in the Methods (p. 25, line 16-25; p.26, line 1-3).

What was the infection efficiency?

We have not measured the infection efficiencies of PDGFRA WT or mutants in MCF-10A and Ba/F3 cells in this project. However, we have measured the infection efficiency of vector carrying GFP and other gene mutations when developing the transient transduction assay. We have extensively explored the effects of mutations in a gene on expression both at the RNA and protein level in citation 19 noted above ([http://www.cell.com/cancer-cell/fulltext/S1535-6108\(18\)30021-7](http://www.cell.com/cancer-cell/fulltext/S1535-6108(18)30021-7)). Thus it is unlikely that there was a significant difference in the infection efficiencies of different mutations of same gene. As shown in citation 19, the transformation activity of a mutation was not correlated with virus infection efficiency or with the expression of the transgene, which have now stated in the Results (p. 6, line 15-16).

It is not clear whether the authors produced several independent cell lines for each of the key mutants.

We have produced several independent stable cell lines in both MCF10A and Ba/F3. We have now clarified this point in the in the Results (p.9, line 6-8, line 22-24; p. 20, line 14-16).

- In figure 1a and suppl table 1, the location (extracellular/kinase/other) of all reported variants is shown. Since the present study suggests that a large part of these variants may not confer any advantage (likely passenger mutations), it would be interesting to plot the number of confirmed gain-of-function drivers in each category and each tumor type.

The variants included in figure 1a and Supplementary table 1 are all PDGFRA variants identified in the GENIE dataset, which included PDGFRA variants that were not included in this study. We have now included a graph showing the number of activating mutations verified in this study in each domain and each tumor type identified from the TCGA, GENIE, and MDACC datasets in supplementary figure 1b. Description of the graph is in p.6, line 17-19 and p.8, line 6-8.

Also, what is the total number of analyzed samples, including wt, for each tumor type? We have now provided the total number of samples per tumor type in Table 1 and Supplementary table 1. The total number of analyzed samples in the GENIE dataset is clarified in p. 8, line 3.

- In suppl figure 1, an arbitrary threshold was used. Did the authors perform a statistical analysis to validate the results?

As suggested by the reviewer, we have now analyzed the results with Dunnett's multiple comparisons test. Mutations with significant gain-of-function when compared to wild-type are indicated in the graphs (Fig 1a and Supplementary Fig 1a) and adjusted *p*-value is provided in the manuscript (p. 6, line 9-14).

- A striking result of this study is the resistance of the Y288C mutant to TKI (figure 6). It would be important to confirm these key results using Ba/F3 cell lines as an independent model.

The resistance of Y288C to TKI has now been confirmed in Ba/F3 stable cell lines and added in Supplementary Fig 7.

- Several studies have reported that kinase transgenes may acquire additional unwanted mutations, particularly when using retroviral vectors (Watanabe-Smith et al, Oncotarget, 2017, Vol. 8, pp: 12596-12606). The authors should therefore sequence the transgene in stable cell lines. This is particularly important for the drug sensitivity tests shown in figure 6, because additional mutations may be responsible for the observed resistance to TKI.

We are aware of this analysis and have sequenced all transgenes routinely. Indeed, all stable cell lines used in presented data had been sequenced and expressed only the mutation of interest. We have now elucidated this point in the Methods (p. 26, line 8-9).

- It is difficult to conclude from Suppl fig 7 on the presence or absence of a disulfide bridge to stabilize the Y288C dimer.

In figure 3c, we have shown that the dimerized Y288C was largely decreased by reducing agent, suggesting the presence of disulfide bond between Y288C dimers (p. 15, line 7-10). However, the disulfide bond formed by the unpaired cysteine in Y288C is not the only factor to stabilize the dimer since PDGFRA dimers were also detected in cells expressing Y288H, which is absence of unpaired cysteine for the disulfide bond formation (Supplementary figure 6a). All that can be stated is that a disulfide bond plays a role. It does not have to be an interchain bond but could be intrachain. We have now revised the text and clarified the concept (p. 16, line 10-14).

- Suppl figure 5 shows constitutive signaling in Ba/F3 cells expressing Y288C. Because the level of residual signaling is highly variable, depending on the starvation conditions, a control cell line is needed (Ba/F3-empty vector and/or WT).

We were not able to establish Ba/F3 stable cell lines expressing WT or Ba/F3-empty vector as they do not survive without the IL-3. Thus, while we agree that this would be a good control, it cannot be done due to the way in which the Ba/F3 cells lines are established.

It is also unclear whether this "stable" cell line is grown in the absence of IL-3. We have stated that the Ba/F3 stable cell lines were grown in the absence of IL-3 in the Methods (p. 26, line 1-3). We have now further clarified this point in the Results (p. 8, line 21-23).

- Bahlawane et al published an interesting study on the same topics in Cell Communication and Signaling (2015, v13:pp21-), which should be cited. The suggested study is now cited as citation 35 in p. 10, line 22-23 and p. 18, line 21-22.

Reference 51 (a conference abstract) has not been confirmed by a full publication. We have now removed reference 51 and the corresponding information.

Referring to Reviewer #2's comments:

Major Comments

1) Of the 16 mutations tested, 3 were associated with 'activating' PDGFRA activity. This raises the possibility that the remaining 13 may be bystander or passenger mutations. I would like to see comprehensive analysis, guided by the functional data, to assess if there are any sequence features (eg location, conservation, crystal structure, rate of recurrence, etc) that might have predicted that the 13 are different from the 3. Such information may prove important in improving DNA sequenced based bioinformatics algorithms to discern true drivers from passengers.

Based on reviewer's suggestion, we have now included the information of the location of the mutations on the 3D crystal structure and also the rate of recurrence in Table 1. Also, we have analyzed and compared the rankscore of the pathogenicity likelihood and gene conservation between activation and non-activating mutations (Supplementary Table 1 and Supplementary Fig 1). Description and interpretation on the results are now included in p.5, line 9, 12-25; p. 6, line 20-25; and p. 7, line 1-14.

2) For the MCF10A experiments, it is not clear if the authors tested all 16 mutations, or just the 3 that were positive in the BA/F3 model. If the latter, it would be interesting to test all 16 mutations, as there is a possibility that some of the 'negative 13' may show activity in a different cell type.

We have tested the activity of all 16 mutations in MCF10A and the result is shown in

Supplementary Fig 1. The “negative 13” that were negative in the Ba/F3 assay remained negative in the MCF-10A cell assay.

3) To what extent are the neo-morphic feature of Y288C (including Golgi/ER retention) a consequence of very high levels of stable expression? Conceptually, it remains possible that if wild-type PDGFRA is also expressed to the same amount seen in the Western blots, then this would result in Golgi trapping, signalling etc. This could be addressed either by titrating the amount of Y288C to levels comparable to WT PDGFRA, and then comparing the two; (or in a more challenging fashion, CRISPR engineering PDGFR to harbour the Y288C mutation). This Reviewer’s point is that if it is indeed a consequence of ‘simple high overexpression’, then there is a concern about the in vivo relevance of the study, given that the results are based cell line transfections that usually result in non-physiological levels of overexpression anyway.

This is a potentially important point. We thus titrated the infection using 3 different viral titers. Importantly independent of viral titer, the amount of Y288C protein were higher than that of WT, and dominantly high mannose glycosylated in all 3 infections (Supplementary figure 4b). Further, we isolated single cells clones in an attempt to obtain cells with comparable amount of Y288C to PDGFRA WT. However, in all of the isolated clones (13 clones of PDGFRA WT and 17 clones of Y288C), the Y288C protein were higher than WT and dominantly expressed as high mannose glycosylated with similar amounts in the different clones, which was independent of the mRNA expression in both WT and Y288C single cell clones (Supplementary figure 4c and d). Thus high RNA levels of PDGFRA WT was not sufficient to induce mislocalization or increased in protein levels. Thus the high level of Y288C appears due to higher protein stability of high mannose glycosylated receptor (Fig 2e) rather than to overexpression per se.

4) I would like to see the Discussion expanded with more implications for cancer therapy. In an age where there is often a rush to treat patients based on sequence-based mutation data alone, it is conceivable that patients with Y288C mutations might be treated with PDGFRA inhibitors, which by the investigator’s data is precisely something *not* to do. Hence, this study is a good cautionary tale about the potential pitfalls of relying on genomic results in isolation.

As suggested by the reviewer, we have now elaborated the potential clinical implications of our findings on cancer therapy in the Discussion (p.24, line 9-14 and 18-21).

Minor Comments

1) More info could be provided about the 16 mutations tested. Were they selected using some criteria, or do these 16 represent the totality of novel PDGFRA mutations identified in TCGA/GENIE? MDACC? Are these 16 PDGFRA mutations associated with

any specific tumor types? Are they recurrent or singletons?

We selected mutations that have not been studied or characterized. This has now been clarified (p. 5, line 4-5). The 16 selected PDGFRA mutations were mostly identified in cutaneous melanoma (7 mutations), glioblastoma (6 mutations), and colorectal adenocarcinoma (5 mutations) (p. 5, line 6-8). We have now included the frequencies of each mutations from the TCGA, GENIE, and MDACC datasets, respectively, in Table 1. Eleven out of 16 mutations are recurrent mutations (p. 5, line 9).

2) Data showing the initial selection of the 3 Ba/F3 positive mutations needs to be shown as part of Main Figure 1, and compared to the other 13 mutations.

As suggested, we now show the result of the transient assay of Ba/F3 in main figure 1.

3) Is the following a typographical error? The first part of the sentence seems to contradict the second – “In MCF10A and Ba/F3 cells stably expressing PDGFRA WT (we were unable to establish a stable cell line expressing the PDGFRA WT in Ba/F3 which was likely due to the low transformation activity)”. Can Ba/F3 cells be stably transfected or not?

We have now revised the text for clarity (p. 8, line 23-25).

4) Is it possible to remine some of the primary tumor data (if it is available) to ask if tumors with the activating PDGFRA mutations also have high expression of PDGFRA in the same tumor?

Unfortunately, the primary tumor is not available for the majority of the cases. Further, most of the activating mutations were not found in the TCGA dataset which has protein expression data. Thus we were not able to determine whether the activation PDGFRA mutations have high PDGFRA expression in the same tumor.

REVIEWERS' COMMENTS:

Reviewer #1 (Remarks to the Author):

The authors provided satisfactory answers to my comments.

Reviewer #2 (Remarks to the Author):

The authors have addressed my original concerns.